# Synergistic Combination of Aztreonam and Ceftazidime–Avibactam—A Promising Defense Strategy against OXA-48 + NDM *Klebsiella pneumoniae* in Romania

**DOI:** 10.3390/antibiotics13060550

**Published:** 2024-06-12

**Authors:** Ioana Miriana Cismaru, Maria Cristina Văcăroiu, Elif Soium, Tiberiu Holban, Adelina Maria Radu, Violeta Melinte, Valeriu Gheorghiță

**Affiliations:** 1Agrippa Ionescu Clinical Emergency Hospital, 011356 Bucharest, Romania; ioana-miriana.cismaru@rez.umfcd.ro (I.M.C.); elif.soium@rez.umfcd.ro (E.S.); holban.tiberiu@dcti.ro (T.H.); maria-adelina.cosma@drd.umfcd.ro (A.M.R.); violeta.melinte@umfcd.ro (V.M.); valeriu.gheorghita@umfcd.ro (V.G.); 2Faculty of Medicine, Carol Davila University of Medicine and Pharmacy, 020021 Bucharest, Romania

**Keywords:** ceftazidime–avibactam, aztreonam, synergism, *bla*NDM, *bla*OXA-48, carbapenem-resistant *Klebsiella pneumoniae*

## Abstract

With the increasing burden of carbapenem-resistant *Klebsiella pneumoniae* (CR-Kp), including high rates of healthcare-associated infections, treatment failure, and mortality, a good therapeutic strategy for attacking this multi-resistant pathogen is one of the main goals in current medical practice and necessitates the use of novel antibiotics or new drug combinations. Objectives: We reviewed the clinical and microbiological outcomes of seven patients treated at the “Agrippa Ionescu” Clinical Emergency Hospital between October 2023 and January 2024, aiming to demonstrate the synergistic activity of the ceftazidime–avibactam (C/A) plus aztreonam (ATM) combination against the co-producers of *bla*NDM + *bla*OXA-48-like CR-Kp. Material and Methods: Seven CR-Kp with *bla*NDM and *bla*OXA-48 as resistance mechanisms were tested. Seven patients treated with C/A + ATM were included. The synergistic activity of C/A + ATM was proven through double-disk diffusion in all seven isolates. Resistance mechanisms like KPC, VIM, OXA-48, NDM, IMP, and CTX-M were assessed through immunochromatography. Results: With a mean of nine days of treatment with the synergistic combination C/A + ATM, all patients achieved clinical recovery, and five achieved microbiological recovery. Conclusions: With the emerging co-occurrence of *bla*OXA-48 and *bla*NDM among *Kp* in Romania, the combination of C/A and ATM could be a promising therapeutic option.

## 1. Introduction

In recent years, some resistant and hypervirulent strains of *Kp* have obtained notable pathogenic properties, standing out as some of the most common microorganisms in serious and even fatal diseases [1].

According to a meta-analysis including 35 original studies addressing colonization with carbapenem-resistant *Kp,* which evaluated 37,661 patients from 18 countries between 2010 and 2021, the prevalence of colonization varies by location and ranges from 0.13 to 22%, with a pooled prevalence of 5.43% (3.73–7.42) [2].

Based on a report from the European Antimicrobial Resistance Surveillance Network between 2016 and 2020, the general burden of multidrug-resistant infections was estimated to be the highest in Greece, Italy, and Romania, with the highest individual yearly estimates occurring in Romania from 2018 to 2019 [3,4].

In parallel with the COVID-19 pandemic, despite all the prevention measures taken in Romania, another “silent pandemic” has been ongoing. The high use of antimicrobials in hospitalized patients has increased the rate of colonization with resistant microorganisms and favored the development of XDR bacteria, especially among oncologic patients and the seropositive population. Romania has jumped to the second position among EARS Net member states regarding carbapenem-resistant isolates [5,6,7,8].

According to Romanian national statistics for 2021, among 531 *Kp* strains, when 227 CR-Kp isolates were tested for colistin sensitivity, 36.7% were colistin-resistant. The increased circulation of pathogenic *bla*NDM *Kp* and the high percentage of colistin-resistant CR-Kp in the absence of therapeutic rescue regiments such as cefiderocol makes it imperative to devise clear strategies for prevention and treatment [9,10].

Starting in March 2022, data have shown an increasing incidence of *bla*NDM-1 and *bla*NDM-1/*bla*OXA-48-producing *Kp* in Germany and Denmark, coinciding with the outbreak of war in Ukraine. In 2021, Ukraine also reported high antimicrobial resistance in Gram-negative bacteria, including CR-Kp, in 64% of tested strains. In addition to increasing trends in travel and population mobility, the above epidemiological data must be taken into consideration regarding the carriage of multidrug-resistant organisms for the better implementation of prevention and control measures [11].

In times of constant epidemiological change, it is crucial to understand the pathogenic mechanism of *Kp.* Carbapenem hydrolysis is mainly mediated by enzymes classified as Ambler class A (KPC), class B (VIM, IMP, and NDM), and class D (OXA-48). The growing spread of NDM has created an urgent need to identify therapeutic options, as the effect of β-lactamases inhibitors has significantly decreased [12,13].

The EMA-approved antibiotic C/A has a bactericidal effect against Ambler classes A and C, and some of the D class isolates like *bla*OXA-48 CR-Kp, but it has no effect on Ambler class B isolates, so this combination does not affect metallo-β-lactamases (MBLs), including *bla*NDM [14,15]. When adding ATM to C/A, the inhibitory activity against carbapenem-resistant Enterobacteriaceae can be obtained owing to the stability of ATM against MBLs, as can the protection of avibactam against aztreonam-hydrolyzing enzymes like ESBLs and AmpC, expanding the effect of ATM + C/A on *bla*NDM and *bla*OXA-48 co-producers [16,17]. These enzymes are often associated with carbapenemases. 

In the following case series study, we evaluated the clinical and microbiological outcomes of seven patients who received a combination of C/A plus ATM for co-producing *bla*NDM and *bla*OXA-48 *Kp* isolates.

## 2. Results

At our hospital, we experienced an increasing trend in CR-Kp infections, noting a growing number of XDR strains. Over 4 months, between October 2023 and January 2024, we identified 43 CR-Kp strains isolated from multiple sites.

In our case series, we selected seven patients admitted during this period with positive cultures for CR-Kp producing *bla*NDM, *bla*OXA-48, and CTX-M. Regarding their distribution among hospital wards, three of them were admitted to the ICU/surgical ward after surgical procedures, and four of them were admitted to medical wards.

All seven subjects were adult males aged between 57 and 84, with a mean of 66 years. No CR-Kp producing *bla*NDM and *bla*OXA-48 was isolated in any female patients. More than half of the subjects had prior contact with medical settings, as four out of the seven patients were hospitalized in the last 3 months, and four had undergone surgical procedures in the last 6 months. Regarding the risk factor of developing antimicrobial resistance, six out of the seven were treated with at least one course of antibiotics, and three were exposed to carbapenems in the last 3 months. The group characteristics show patients with multiple comorbidities, especially neurological, oncological, cardiac, and urological. The Charlson index score was between two and seven. Four patients tested positive for bacterial colonization with different MDR bacteria at a routine screening, including XDR *Kp*, VRE, MRSE, CPE, and ESBL. All the patients were exposed to invasive procedures, urinary catheterization was performed in all cases, and three patients had a CVC placed and were under orotracheal intubation during surgical procedures (Figure 1, Figure 2, Figure 3, Figure 4 and Figure 5).

None of the patients met the criteria for septic shock according to their SOFA scores; two were diagnosed with malignancies and underwent extensive abdominal surgery during hospitalization, but none received immunosuppressant therapy.

The specimens were collected from three different sites: four urinary, two intra-abdominal, and one blood culture. Only one subject had bacteriaemia, and the probable source was obstructive pyelonephritis, even though the urinary culture was negative. Two patients had symptomatic urinary infections, and two had asymptomatic urinary infections. Two intra-abdominal cultures were multi-bacterial, including CR-Kp, and were collected intraoperatively. In one case, control cultures could not be performed, as the patient did not undergo a second surgical procedure.

All the CR-Kp isolates were *bla*NDM, *bla*OXA-48, and CTX-M producers, and all had an MIC of >16 mg/L for meropenem and C/A and an MIC of >64 mg/L for ATM. Cefiderocol resistance was proven in one specimen and colistin resistance in six out of the seven. Fosfomycin resistance was proven in five out of the seven isolates (two were not tested). Furthermore, high resistance to tigecycline was noticed. The synergistic activity of C/A + ATM was demonstrated in all seven isolates (Table 1).

All patients received combination therapy with 2.5 g C/A every 8 h and 2 g of ATM every 8 h, infused over 3 h with no other associated antimicrobial agent. In one patient with end-stage kidney disease, higher 1.25 g doses of C/A q12h + 2 g of ATM q12h were administered at an eGFR of 20–30 mL/min, with intermittent dialysis sessions, to ensure higher serum concentration. We could not determine serum concentration measurements at our hospital. No other renal adjustments were necessary in the other cases. The treatment was generally well tolerated, with no major adverse reactions.

The treatment duration was between 7 and 14 days, with a mean of 9 days. All patients achieved clinical recovery, and five achieved microbiological recovery. In one case, control cultures could not be performed, and another patient had a positive culture at the end of treatment, but both subjects had good clinical and inflammatory responses. All urinary control cultures were negative on day 7 of treatment. The minimum hospital stay was 10 days, and two patients were hospitalized for more than 30 days owing to unrelated affections. On day 30, mortality was zero, and no re-admissions were registered on account of CR-Kp infections (Table 2).

## 3. Discussion

Modern medicine is threatened by growing antimicrobial resistance, especially among Gram-negative microorganisms, where resistance to β-lactams is most often mediated by β-lactamases. Throughout much of Europe, Northern Africa, and the Middle East, the OXA-48 enzyme has proliferated, becoming the most prevalent carbapenemase [18].

The first data regarding the presence of carbapenemase-producing Enterobacteriaceae (CPE) isolates in Romania were presented in a review by Cantón et al. in 2012, based on unpublished data [19]. In recent years, there has been an increase in reports of *Kp* isolates harboring two carbapenemases [20,21,22,23]. There is a growing number of strains that produce multiple carbapenemases in other enterobacterial species [24].

A total of 832 genome sequences of CR-Kp strains co-carrying two of the five major carbapenemase genes from 105 countries from 1980 to 2022 were retrieved from the NCBI GenBank database. The strains that co-carried *bla*NDM and *bla*OXA-48-like genes accounted for 665 out of the total (79.9%), ranking first, and strains that co-carried *bla*KPC and *bla*NDM genes accounted for 103 (12.4%), ranking second [25].

Despite our small amount of data from this small group of patients, our case series shows the importance of studying the mechanisms underlying the co-occurrence of multiple carbapenemase types and indicates a possible upward trend for CR-Kp co-producing *bla*OXA-48 and *bla*NDM in Romanian hospitals, calling for further research regarding the prevalence of highly resistant strains in our country.

In the current scenario, we aimed to highlight that having good knowledge of the epidemiological context and molecular mechanisms of CR-Kp can lead to appropriate therapeutic choices. It is crucial to understand which antibiotics are effective against each type of carbapenemase to tailor therapy and improve clinical outcomes. Our results showing CR-Kp strains co-carrying three mechanisms of antimicrobial resistance (*bla*NDM, *bla*OXA 48, and CTX-M) emphasize the need to improve the efficiency and accuracy of standard AMR data analysis and reporting workflows.

Starting in 2010, a few published Romanian studies have signaled the presence of *bla*OXA-48 and *bla*NDM *Kp* isolates in patients admitted to hospitals from different geographic areas. The first study was conducted between November 2013 and April 2014, and it described the local distribution of *bla*NDM-1, *bla*OXA-48, and *bla*OXA-181 in nine hospital isolates from central Romania, whereas the second one showed the distribution of CPE in two hospitals in Bucharest between 2011 and 2012 [26,27,28,29].

Between October 2023 and January 2024, we identified 43 CR-Kp strains, from which 7 strains were tested using immunochromatography, revealing *bla*NDM and *bla*OXA 48 enzyme co-production. All isolates demonstrated resistance to C/A, monobactams, and carbapenems, and one showed resistance to cefiderocol.

A six-month survey conducted between November 2013 and April 2014 at eight Romanian hospitals presented the first characterization of CR-Kp. It showed that among 75 non-susceptible isolates, 65 were carbapenemase producers. The most frequently identified genotype was *bla*OXA-48 (*n* = 51 isolates). Eight isolates were positive for the *bla*NDM-1 gene, four had the *bla*KPC-2 gene, and two were positive for *bla*VIM-1 [30].

Compared with previous studies, our findings strongly suggest that there is an underestimation of the prevalence of co-producing *bla*NDM and *bla*OXA-48 *Kp* isolates, and further research is a pressing priority, especially on the use of novel antibiotic combinations. The current guidelines recommend C/A plus ATM or cefiderocol alone for *bla*NDM-producing strains and C/A alone for *bla*OXA-48-producing strains for extra-urinary tract infections. There are no recommendations for when two or more resistance mechanisms are detected within the same *Kp* strain [31].

In our survey, all seven patients were treated with a combination of C/A and ATM. All achieved clinical recovery, and five achieved microbiological recovery with zero mortality and no re-admission owing to CR-Kp. We noticed no severe adverse events, although the safety of this combined drug therapy is not well established owing to a lack of data. However, the small number of patients enrolled and the short period of monitoring are limitations in this regard. Consistent with our promising outcomes, another published review highlighted the favorable outcomes of ten patients treated with C/A plus ATM in Spain, achieving clinical success in six of the subjects. All infections were caused by the hypervirulent strain KP-HUB-ST147, which is resistant to ATM and C/A [32].

Other data have demonstrated the success of the C/A and ATM combination in two different patients, the first with a hip arthroplasty site infection caused by an XDR NDM-1-producing strain of Enterobacter cloacae and the second with a suppurated thrombophlebitis and persistent bacteremia caused by an OXA-48/NDM-1-producing *Kp*, both pathogens with proven in vitro synergistic activity [33,34].

Our case series suggests that the C/A and ATM combination could be successfully used for patients with limited therapeutic solutions due to antimicrobial resistance or in countries with limited accessibility to cefiderocol. It is the only β-lactam active against MBL-producing *Kp*, but strains with reduced susceptibility to this compound are already emerging despite their short-term use [35].

Using our unpublished data, we identified a significant percentage of cases with resistance to cefiderocol, proven using the disk diffusion method (according to EUCAST criteria 2023/2024).

Several findings in the literature suggest a correlation between NDM-type β-lactamases and important increases in cefiderocol MIC, demonstrated by the introduction of NDM genes in isogenic mutants [35,36,37] and the much higher prevalence (42–59% in some cohorts) [38,39,40] of cefiderocol nonsusceptibility in *bla*NDM-producing isolates from clinical practices. Moreover, NDM expression appears to facilitate the emergence of cefiderocol resistance through additional mechanisms (such as mutations in siderophore receptors) [41].

An alarming resistance to cefiderocol is likely to develop over time owing to the increasing number of *bla*NDM strains, as seen in our epidemiological findings. Cross-resistance among ceftazidime–avibactam, ceftolozane–tazobactam, and cefiderocol has been reported and is associated with KPC variants in *K. pneumoniae* [24,41] or AmpC variants in *Enterobacter* spp. and *P. aeruginosa* [42].

In Romania, cefiderocol is approved by the National Agency of Medicine and Medical Devices but is hardly accessible. Other compounds such as colistin, fosfomycin, tigecycline, minocycline, and eravacycline could have significant toxicity, or these compounds may achieve poor concentrations in some infection sites. CR-KP is also developing resistance to these agents at an alarming rate [37].

The combination aztreonam–avibactam has been studied given that ATM is stable against metallo-β-lactamases and avibactam can protect against aztreonam-hydrolyzing enzymes, including ESBLs and AmpC β-lactamases, which are often coproduced by these highly resistant variants [43].

A study conducted on 422 patients with complicated intra-abdominal infections (CIAI) or HAP/VAP comparing aztreonam–avibactam versus meropenem–colistin showed the first combination was effective, displaying similar efficacy and good outcomes at 28-day all-cause mortality. A systematic review including 35 in vitro and 18 in vivo studies on the combination aztreonam–avibactam for MBL producers revealed clinical resolution within 30 days for 80% of infected patients [44,45].

Our study is limited by the small number of patients in the case series. In our region, resistance genes are not routinely tested for, and the first option for clinicians is treatment based on little epidemiological data and clinical and microbiological indicators. All these factors make medical decisions and treatment options difficult regarding CR-Kp co-producing *bla*OXA-48 and *bla*NDM, particularly when there is limited access to other active novel antibiotics.

## 4. Materials and Methods

### 4.1. Setting and Patients

In this observational, retrospective, noninterventional study, we reviewed the outcomes of 7 patients who were admitted to the Clinical Emergency Hospital “Agrippa Ionescu”, Bucharest, Romania, between October 2023 and January 2024. We selected all patients with CR-Kp producing *bla*NDM + *bla*OXA-48 isolated in cultures; with demonstrated resistance to C/A, monobactams, and carbapenems; and who were treated with the synergic combination of C/A + ATM. An infectious diseases physician was consulted in all cases, and all patients consented to the treatment. Both antibiotics are approved for use by the Romanian National Agency of Medicines and Medical Devices.

The specimens were acquired from different sites: urinary tract, bloodstream, and intra-abdominal collections. We included patients with bacteriaemia, symptomatic site infections, or asymptomatic site infections. Symptomatic site infections were described as positive cultures with organ-specific symptoms, signs of generalized infection, or inflammatory responses. Asymptomatic site infections were defined as positive cultures with inflammatory responses and without any clinical manifestations.

To evaluate inflammatory responses, we monitored C-reactive protein levels. Clinical recovery was defined as the remission of systemic or generalized clinical signs of infection (fever, shivers, dysuria, and abdominal pain) associated with a significant decrease in the inflammatory response (obtaining a C-reactive protein level below 10 mg/L) during the follow-up period. Microbiological recovery was achieved when negative cultures were obtained. Patients were followed up for 30 days after treatment. Recurrence was defined as a new positive culture with or without clinical signs and symptoms of infection after the first clinical or microbiological success.

### 4.2. Microbiology

A total of 7 specimens were obtained with the described characteristics. Bacterial identification and antibiotic susceptibility testing were performed with Vitek2 compact (BioMerieux, Mercy-l’Étoile, France), and the results were interpreted in accordance with the EUCAST 2023 criteria. The breakpoints used for defining resistance for meropenem and C/A were an MIC of >8 mg/L; for ATM, an MIC of >4 mg/L; and for colistin, an MIC of >2 mg/L. Cefiderocol sensitivity was tested in vitro via disk diffusion (R < 23 mm) as per the EUCAST 2023 criteria. A double-disk diffusion test was performed for all isolates to determine the effect of C/A and ATM in combination with each other against the resistant microorganism. Mueller–Hinton agar (BioMerieux, Mercy-l’Étoile, France) was inoculated with a saline suspension from a fresh culture of the isolated microorganism adjusted to 0.5 McFarland turbidity standard. The synergism was tested by placing 2 antibiotic disks of C/A (10/4 μg) CZA 14 and ATM 30 μg (Oxoid, Hampshire, UK) 20 mm from each other (center to center). After 16 to 20 h of incubation, a synergistic effect was present in the two antibiotics, and an inhibition zone formed between the disks. Molecular resistance mechanisms including KPC, VIM, OXA-48, NDM, IMP, and CTX-M were assessed through immunochromatographic direct detection with rapid NG-Test Carba-5 (NG Biotech, Guipry-Messac, France).

### 4.3. Analyzed Variables

The data were obtained from a microbiology laboratory database and the patients’ electronic records. In addition to demographics, other risk factors included the following: admission to a ward with a high risk of acquiring MDR infection or colonization; previous hospitalizations in the last 3 months; surgery in the last 6 months and the type of surgery; presence of indwelling devices or invasive maneuvers during hospitalization; type and number of underlying comorbidities; Charlson index; colonization with MDR pathogens at admission on the wards or ICU; and prior antibiotic and carbapenem exposure in the last 3 months (Figure 1, Figure 2, Figure 3, Figure 4 and Figure 5).

We analyzed the specimen collection site, susceptibility to antibiotics with MIC testing, beta-lactamase production testing, and Ambler class classification, as seen in Table 1.

We monitored the duration of treatment; the length of hospitalization; clinical and microbiological recovery; re-admission by day 30 after discharge owing to infection with CR-Kp or other non-related affections; and 30-day mortality by any cause (Table 2).

### 4.4. Treatment

All patients received combination therapy with 2.5 g C/A every 8 h and 2 g ATM every 8 h in prolonged infusion over 3 h according to the IDSA 2023 Guidance on the Treatment of Antimicrobial-Resistant Gram-Negative Infections, except for 1 patient with prior end-stage kidney disease (included in a dialysis program). In this case, higher doses were preferred to creatinine-clearance-adjusted doses to ensure a higher serum concentration, as the antibiotic serum concentration measurement could not be performed at our hospital. None of the patients received any other antibiotics simultaneously.

## 5. Conclusions

In the current scenario, we aimed to highlight the fact that good knowledge of epidemiological data and molecular resistance mechanisms leads to appropriate therapeutic choices.

Our clinical data suggest that combination therapy with C/A plus ATM is a safe and efficient therapeutic option for treating severe infections caused by CR-Kp producing MBLs associated or not associated with OXA 48 and CTX-M, and this should be considered a first-line therapeutic option in the absence of other β-lactam or non-β-lactam antibiotics.

The optimal use of old or novel antibiotics or a synergistic combination of antibiotics represents an effective strategy against AMR development and improves prognosis.

Infection prevention and control are also critical for reducing the spread of microorganisms producing MBLs in healthcare systems.

## Figures and Tables

**Figure 1 antibiotics-13-00550-f001:**
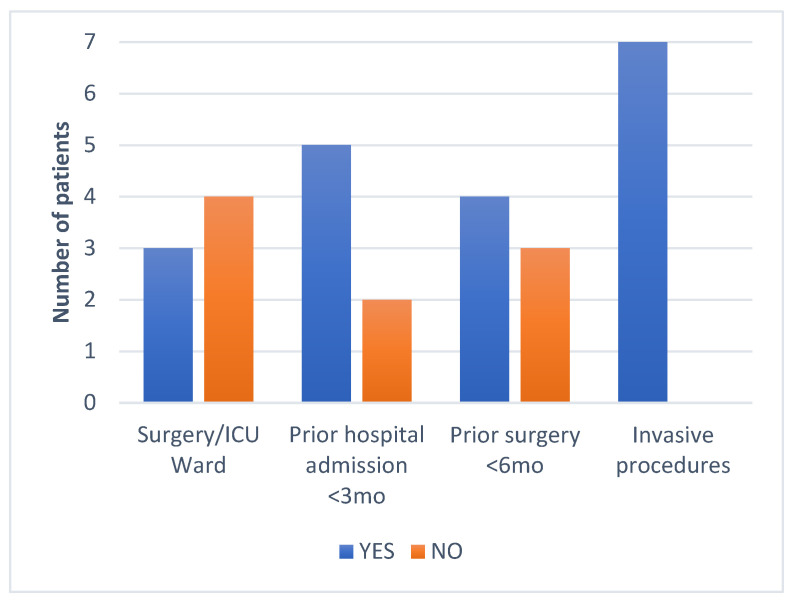
Patient medical history.

**Figure 2 antibiotics-13-00550-f002:**
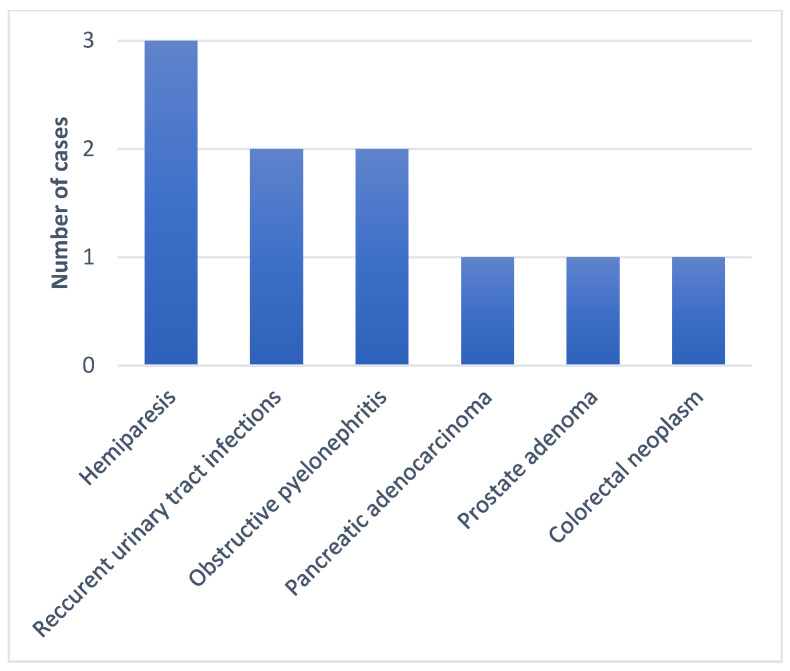
Underlying diseases.

**Figure 3 antibiotics-13-00550-f003:**
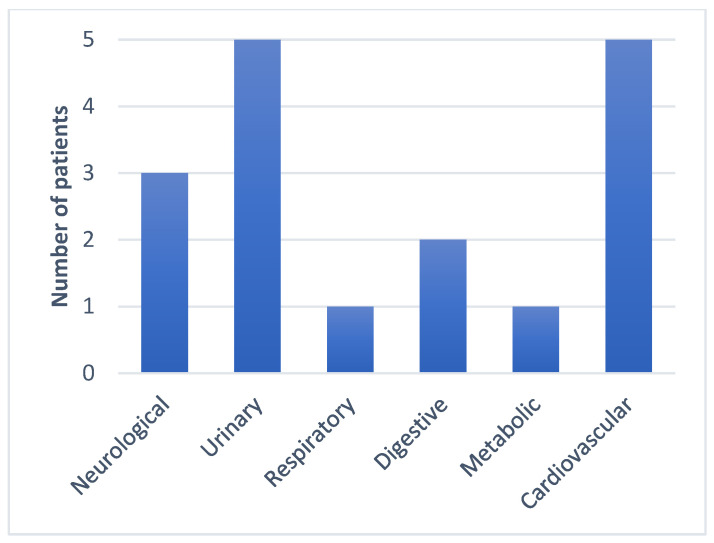
Associated comorbidities.

**Figure 4 antibiotics-13-00550-f004:**
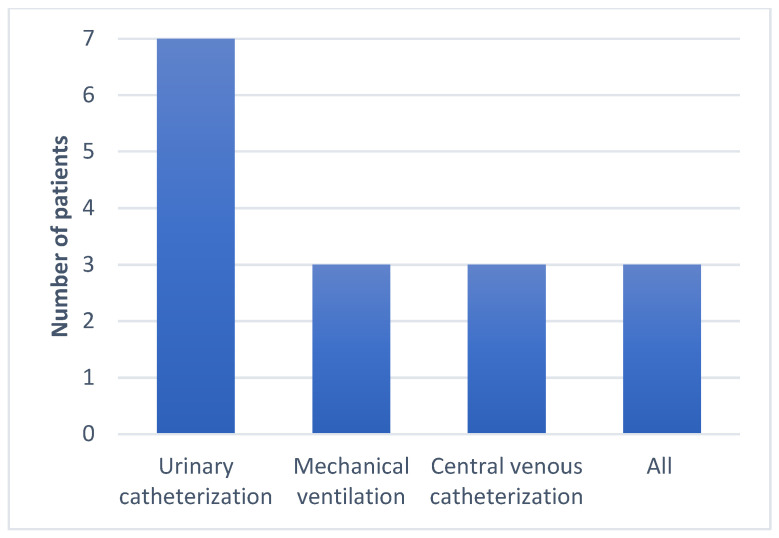
Type of invasive procedures.

**Figure 5 antibiotics-13-00550-f005:**
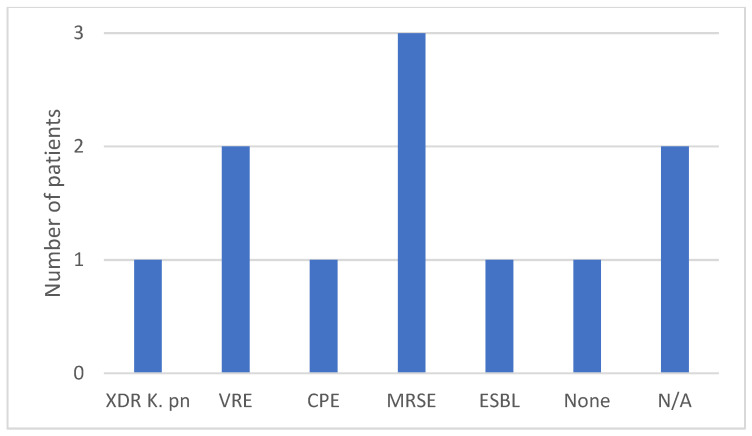
Previous bacterial colonization and type of bacteria.

**Table 1 antibiotics-13-00550-t001:** Bacterial characteristics; source of infection; type of carbapenemase; susceptibility profile.

Patient	Previous Antibiotic/Carbapenem Exposure	Site	Ambler Class	β-Lactamases	C/A (MIC)	ATM (MIC)	SYN	Cefiderocol	Colistin (MIC)	Meropenem (MIC)	Tigecycline	Fosfomycin
1	No/No	Urinary	A, B + D	NDM + Oxa-48 + CTX-M	R (>16 mg/L)	R (>64 mg/L)	Yes	S	R (>16 mg/L)	R (>16 mg/L)	S	R
2	Yes/No	Urinary	A, B + D	NDM + Oxa-48 + CTX-M	R (>16 mg/L)	R (>64 mg/L)	Yes	S	S (0.5 mg/L)	R (>16 mg/L)	R	R
3	Yes/Yes	Bacteriemia	A, B + D	NDM + Oxa-48 + CTX-M	R (>16 mg/L)	R (>64 mg/L)	Yes	S	R (>16 mg/L)	R (>16 mg/L)	I	-
4	Yes/Yes	Biliary	A, B + D	NDM + Oxa-48 + CTX-M	R (>16 mg/L)	R (>64 mg/L)	Yes	S	R (>16 mg/L)	R (>16 mg/L)	R	-
5	Yes/No	Urinary	A, B + D	NDM + Oxa-48 + CTX-M	R (>16 mg/L)	R (>64 mg/L)	Yes	R	R (>16 mg/L)	R (>16 mg/L)	R	R
6	Yes/No	Urinary	A, B + D	NDM + Oxa-48 + CTX-M	R (>16 mg/L)	R (>64 mg/L)	Yes	S	R (>16 mg/L)	R (>16 mg/L)	I	R
7	Yes/Yes	Peritoneal	A, B + D	NDM + Oxa-48 + CTX-M	R (>16 mg/L)	R (>64 mg/L)	Yes	S	R (>16 mg/L)	R (>16 mg/L)	R	R

C/A—ceftazidime–avibactam; ATM—aztreonam; SYN—synergism between ceftazidime–avibactam and aztreonam; R—resistant; S—sensitive; I—intermediate.

**Table 2 antibiotics-13-00550-t002:** Characteristics and outcomes of patients treated with ceftazidime–avibactam and aztreonam.

Patient	Duration of Treatment	Doses (C/A + ATM)	Length of Hospitalization (Days)	Clinical Recovery	Microbiological Recovery	Recurrence at 30 Days	Re-Admission (Other) at 30 Days	Death at 30 Days
1	7	2.5 g q8h + 2 g q8h	12	Yes	Yes	No	Yes	No
2	7	2.5 g q8h + 2 g q8h	10	Yes	Yes	No	Yes	No
3	10	2.5 g q8h + 2 g q8h	>30	Yes	Yes	No	Yes	No
4	7	2.5 g q8h + 2 g q8h	28	Yes	N/A	No	No	No
5	14	2.5 g q8h + 2 g q8h	14	Yes	Yes	No	Yes	No
6	12	1.25 q12h + 2 g q12h	20	Yes	Yes	No	No	No
7	7	2.5 g q8h + 2 g q8h	>30	Yes	No	No	No	No

## Data Availability

The data are contained within the article.

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
