# Peer review of "Synergistic Combination of Aztreonam and Ceftazidime–Avibactam—A Promising Defense Strategy against OXA-48 + NDM Klebsiella pneumoniae in Romania"

_antibiotics, 2024, doi:10.3390/antibiotics13060550_

Round 1

Reviewer 1 Report

Comments and Suggestions for Authors

The authors reviewed the clinical and microbiological outcomes of 7 patients after characterizing the carbapenemases and providing a combination therapy comprising C/A and ATM. 

Comments on the Quality of English Language

None

Reviewer 2 Report

Comments and Suggestions for Authors

Ioana Miriana Cismaru et al. present a very interesting article.

However, the results are presented in a very short and compressed way. I kindly ask the authors to write the results in a much more comprehensive way and illustrate them with additional figures. Furthermore, the discussion should be more scientific. 

To sum up, Dr. Cismaru presents a very important article, which urgently needs major revisions. I'm looking forward to review the article again after revisions. 

Comments on the Quality of English Language

Moderate editing of English language required

Reviewer 3 Report

Comments and Suggestions for Authors

This is a case series of 7 patients with MBL-producing infections treated with ceftazidime-avibactam and aztreonam conbimation in a single hospital in Romania. MBL-producing infections are difficult to treat and ceftazidime-avibactam and aztreonam is a promising new combination for their treatment. Few data exists on the treatment of patients with MBL infections with ceftazidime-avibactam, therefore the topic of this paper is important.

The authors have done a fair effort to present in depth the cases treated in their centre and results are interesting. That said, I do have some concerns regarding the study ethics and methods/results as presented.

- Given the small number of patients in this case series, the tight timeframe and the detail of each case presented in Table 1, it makes it very likely that the patients described in this paper can be identified. The Ethics section of this paper state ": Written informed consent was obtained from all subjects and/or family members involved as part of the hospital admission procedure." It is unlikely this was done prospectively given the protocol date is stated as 01/04/2024 (protocol no. 241564/ 01 April 2024). No REC committee number is stated. The authors should make efforts to anonymise data.

- The method of testing for ceftazidime-avibactam synergy is not analysed in depth. There is no standardised method for testing for ceftazidime-avibactam & aztreonam synergy therefore the exact procedure should be described for readers to understand the findings better.

- The inclusion criteria for the study as described are not clear. Did the patients include bacteraemias or any infection?

Comments on the Quality of English Language

- Use of the English language could be improved. At most cases content can understood but expressions used are uncommon and could use a review of a fluent English speaker. There are some bodies of text where interpretation of the English is very challenging.

EMA-approved antibiotic C/A has a bactericidal efficiency against blaOXA-48 CR-Kp. 61

Though, avibactam has no effect on metallo-β-lactamases (MBLs) and this β-lactam/β-lac- 62

tamase inhibitor combination lacks activity against Ambler class B isolates producing 63

blaNDM enzymes [14,15]. It is hypothesised to add ATM to C/A. This combination has a 64

strong inhibitory activity against CRE, expanding the coverage over blaNDM and blaOXA-48 enzymes [16,17].

Limitations should be kept in the discussion.

Round 2

Reviewer 2 Report

Comments and Suggestions for Authors

The authors have significantly improved the manuscript. Congratulations. As the manuscript is of high scientific relevance, it should be published as soon as possible. 

Author Response

Thank you again very much for taking the time and for the kind review. 

Reviewer 3 Report

Comments and Suggestions for Authors

The authors' revision has not addressed the key issue of patient confidentialty.

Comments on the Quality of English Language

English remains understandble but could use minor improvements in wording.
